# Compound Heterozygous Missense Variants in *RPL3L* Genes Associated with Severe Forms of Dilated Cardiomyopathy: A Case Report and Literature Review

**DOI:** 10.3390/children9101495

**Published:** 2022-09-29

**Authors:** Bibhuti B. Das, Viswanath Gajula, Sandeep Arya, Mary B. Taylor

**Affiliations:** 1Department of Pediatrics, Division of Cardiology, Children’s of Mississippi, University of Mississippi Medical Center, Jackson, MS 39216, USA; 2Department of Pediatrics, Division of Critical Care, Children’s of Mississippi, University of Mississippi Medical Center, Jackson, MS 39216, USA

**Keywords:** familial dilated cardiomyopathy, *RPL3L*, pediatric heart failure

## Abstract

Whole exome sequencing has identified an infant girl with fulminant dilated cardiomyopathy (DCM), leading to severe acute heart failure associated with ribosomal protein large 3-like (*RPL3L*) gene pathologic variants. Other genetic tests for mitochondrial disorders by sequence analysis and deletion testing of the mitochondrial genome were negative. Secondary causes for DCM due to metabolic and infectious etiologies were ruled out. She required a Berlin-Excor left ventricular assist device due to worsening of her heart failure as a bridge to orthotopic heart transplantation. At three months follow-up after heart transplantation, she has been doing well. We reviewed the literature on published *RPL3L*-related DCM cases and their outcomes. Bi-allelic variants in *RPL3L* have been reported in only seven patients from four unrelated families in the literature. *RPL3L* is a newer and likely pathogenic gene associated with a severe form of early-onset dilated cardiomyopathy with poor prognosis necessitating heart transplantation.

## 1. Case Report

The ribosomal protein large 3-like (*RPL3L*) gene plays a role in myoblast growth and fusion [1]. Bi-allelic missense variants in *RPL3L* have been identified with a severe form of early-onset familial dilated cardiomyopathy (DCM) in seven patients in four unrelated families [2,3]. The identified variants of the *RPL3L* gene are rare in the general population and exhibit an autosomal recessive inheritance pattern. In contrast, most known genetic causes of DCM are autosomal and dominantly inherited. *RPL3L* genes encode the 60S ribosomal protein, which is exclusively expressed in cardiac and skeletal muscles [4]. All identified variants have been missense variants predicted to destabilize *RPL3L* binding to the 60S ribosomal subunit [2]. We describe an infant with fulminant DCM, severe acute heart failure, and *RPL3L* pathologic variants that underwent left ventricular assist device support as a bridge to successful orthotopic heart transplantation.

A female infant born at term to a non-consanguineous African American family with healthy parents presented with signs of hypotension and respiratory distress at two months of age. At presentation, physical examination revealed that her heart rate was 170 bpm, respiratory rate was 56 bpm, and she had mild respiratory distress with subcostal retractions. Her cardiac exam showed normal S1 and S2 with an S3 gallop rhythm. There were equal breath sounds in both lung fields with bi-basilar fine crackles. She had hepatomegaly; the liver was 3 cm below the right subcostal margin but no splenomegaly or ascites was observed. No peripheral edema was observed, and pulses were 2+ in all extremities without radiofemoral delay. The capillary refill was 3–4 s. Her initial laboratory results showed severe metabolic acidosis (pH 7.26, pCO_2_ 16 mmHg, HCO_3_- 7.1 mmol/L, anion gap −16 mmol/L with elevated lactate (>16.8 mmol/L), elevated NT brain-type pro-natriuretic peptide (NTpro-BNP) > 70,000 pg/mL, elevated creatinine (58 µmol/L), and elevated liver enzymes (AST 62 U/L and ALT 105 U/L). Her troponin was 364 ng/L, and creatinine kinase was normal (280 U/L). A chest X-ray showed severe cardiomegaly (Figure 1A). Her cardiac evaluation, including an ECG, showed sinus rhythm with nonspecific ST-segment (Figure 1B). An echocardiogram showed a severely dilated left ventricle (LV) (Figure 1C), markedly decreased LV shortening fraction (FS) (13%) (Figure 1D), and normal coronaries. She required mechanical ventilation and milrinone infusion started with a dose of 0.5 µg/kg/min at presentation. Infectious workups for respiratory pathogens (influenza viruses A and B, parainfluenza virus 1–4, respiratory syncytial virus, rhinovirus, adenovirus, enterovirus, metapneumovirus, mycoplasma pneumonia, and chlamydia pneumoniae) were negative. Serologic tests were negative for cytomegalovirus and the Epstein–Barr virus. Polymerase chain reaction analyses in blood for enterovirus, adenovirus, parvovirus B19, and human herpes simplex virus type 6 were negative. Further workups to rule out metabolic causes of dilated cardiomyopathy, including serum amino acids, urine organic acids, and her newborn screening tests were negative.

She was successfully extubated and remained on intravenous milrinone infusion. A trio-based whole exome sequencing (WES) was performed, and we identified a compound heterozygous variant for c. 151 G > A (p.A51T) and c.691 G > T (p.V231F) in *RPL3L*. Her mother was positive for c.151 G > A (p.A51T), and her father was positive for c.691 G > T (p.V231 F). Both parents are heterozygous carriers (Figure 1E). For this test, WES covered 100% of the coding region of the exome at a minimum of 10×. The copy number variants (CNV) analysis was performed, and there was no indication of a clinically relevant deletion or duplication of three or more exons in the data for our patient. We did not find other variants in the sequencing data. Moreover, the mitochondrial genome analysis was negative. As, the *RPL3L* variant by the phenotype driven testing, we presented our data to our multidisciplinary heart failure and heart transplant team for rigorous risk benefit assessment before “actionable” management. We listed our patient for heart transplantation. According to the American College of Medical Genetics (ACMG) and Genomics guidelines for interpreting sequence variants identified, the *RPL3L* variant is likely pathogenic [5]. The details of the classification of a sequence variant are beyond the scope of this paper, and readers could refer to www.clinvar.com (Accessed on 8 July 2022) for details.

Her cardiac function deteriorated while she was on mirlinone continuous infusion. She had a persistently elevated heart rate (170 bpm), elevated NTpro-BNP (increased from 27,000 to 35,000 pg/mL), slow rise in lactate (2.2 mmol/L), and increasing serum creatinine (62 µmol/L), which led to implantation of a 15 mL volume Berlin Heart Excor^®^ left ventricular assist device (LVAD). Her post-LVAD implantation period was uncomplicated. Unfractionated heparin was used as an anticoagulant, and activated partial thromboplastin time was monitored and kept between 60 and 80 s while waiting in the hospital for a donor heart. After four days of Berlin LVAD implantation, a suitable donor heart was available for her, and she underwent successful heart transplantation.

The parents had no history of symptoms and did not present any signs of heart failure, muscle weakness, or arrhythmias. The family investigation revealed that no other family member in the extended family had cardiomyopathy or heart failure. Since the variant *RPL3L* gene associated with DCM and heart failure is a recessive condition, our genetics counselor counseled the family for future pregnancies.

## 2. Discussion

Dilated cardiomyopathy is the most common form of cardiomyopathy and accounts for approximately 55–60% of all childhood cardiomyopathies, and the majority is presented before one year of age. According to the pediatric cardiomyopathy registry database, the incidence of DCM is 1.1 cases per 100,000 person-years, but the incidence was eight times higher (8.3 cases per 100,000 person-years) in infancy [6]. Genetic testing in individuals with DCM with a positive family history of cardiomyopathy identifies causative mutations in only approximately 25% [7]. Currently, if there is no family history and no known genetic mutation is suspected, using WES as a first-line genetic test to identify the underlying etiology for DCM is recommended [8]. In contrast to other types of genetic testing, WES sequences thousands of genes simultaneously rather than analyzing one or a few genes. WES in the parent–offspring trio provides an efficient means to discover DCM’s genetic basis and ultimately enhances molecular diagnostics yields.

Our experience from our previous case reports [3] helped us to send the WES earlier to identify any underlying genetic basis for DCM in the present case. According to the ACMG guidelines for interpreting sequence variants identified, our patient phenotype is specific for a disease with a single genetic etiology, and prior sources recently reported such variants as pathogenic [2,3]. We believe this case is the tip of the iceberg, and unless tested for these rare possible pathologic variants, as in our case, we will miss identifying high-risk patients. The WES test allowed us to identify a heterozygous variant c.151 G > A (p.A51T) and c.691 G > T (p.V231F) in *RPL3L*. The compound heterozygous variant for c. 151 G > A (p.A51T) and c.691 G > T (p.V231F) in *RPL3L* identified in our case is rarely seen in public databases [4]. Therefore, no frequency data for the general population are available. Various in-silicon tools are publicly and commercially available, which help interpret sequence variants identified by WES. The details of the different software programs and algorithms for their prediction are beyond the scope of the present report. Interested readers can refer to the ACMG report for more information [5]. We summarize the genotypes and phenotypes of seven cases of DCM due to heterozygous compound variants of *RPL3L* in five unrelated families, which have been previously reported in Table 1.

The role of the *RPL3L* gene in heart diseases is still poorly understood. Further studies are therefore needed to unravel its role in cardiac pathophysiology. *RPL3L* sequence variants with the missense variant p.Ala75Val and the splice-donor variant c.1167 + 1G > A have also been implicated in atrial fibrillation [9]. Moreover, another variant *c.724C > T: p.R242W* of the *RPL3L* gene has been reported in a 6-year-old child with catecholaminergic polymorphic ventricular tachycardia [10]. The early identification of *RLP3L* association with DCM may help management, informed family counseling, and in the future, personalized clinical management and risk stratification. The presence of the *RPL3L* variant is associated with rapidly progressive heart failures, and heart transplantation is the best option for long-term success.

## 3. Conclusions

Our case report has a similar clinical presentation to other previously reported cases with *RPL3L* variants associated with severe neonatal DCM with rapid decompensation. It supports the likely pathogenicity of the described *RPL3L* variants for neonatal familial DCM. However, further functional studies are needed to analyze the pathomechanisms underlying *RPL3L*-associated DCM. This case report emphasizes the importance of genetic testing, especially of WES for neonatal DCM, as it can identify variants of possible pathologic significance and can provide an opportunity for early clinical management. The ability to promptly diagnose a neonate with a genetically caused DCM helps clinicians in their discussions with the family with respect to various treatment options, including heart transplantation.

## Figures and Tables

**Figure 1 children-09-01495-f001:**
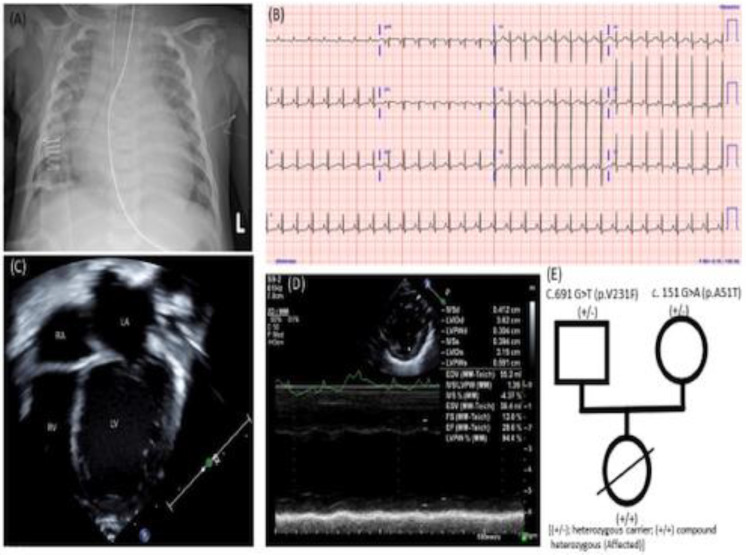
(**A**) Chest X-ray showing severe cardiomegaly at presentation; (**B**) electrocardiogram showing nonspecific ST changes (flat); (**C**) four-chamber view showing a dilated left atrium (LA) and left ventricle (LV); (**D**) m-mode echocardiography displaying depressed LV systolic function (LVSF 13%); (**E**) family pedigree tree.

**Table 1 children-09-01495-t001:** Clinical and genetic findings in seven patients with homozygous variants of the *RPL3L* gene diagnosed using whole exosome sequencing and associated with severe DCM reported in the literature (this table is modified from our earlier publication [3]).

Case No.	Sex	Consanguinity	Age at Diagnosis of DCM	Country of Origin	Other Cardiac Findings	Alive/Dead	Genotypes	Reference
1	F	No	Day1	Germany	PFO PH	Died 21st DOL	c.923A > T (p.Asp308Val) and c.1027C > T (p.Arg343Trp)	#2
2	M	No	Day6	Germany	TR	Died 15th DOL	c.923A > T p.(Asp308Val) and c.1027C > T (p.Arg343Trp)	#2
3	M	Yes	Day75	Colombia	TR, MR RBBB ST-T abnormalities	HT at 6 months Alive At 9 years	c.566C > T (p.Thr189Met) and c.922G > A (p.Asp308Asn)	#2
4	F	No	Day48	Spain	MR ST and T abnormalities	HT at 5 months Alive At 10 years	c.80G > A (p.Gly27Asp) and c.481C > T (p.Arg161Trp)	#2
5	M	No	Day12	Spain	MR, TR. ST and T abnormalities, VSD	Died 30th DOL	c.80G > A (p.Gly27Asp) and c.481C > T (p.Arg161Trp)	#2
6	M	No	Day1	USA	MR ST and T abnormalities	Died 124th DOL	c.1076_1080delCCGTG (p.Ala359Glyfs*4) and c.80G > A (p.Gly27Asp)	#3
7	M	No	Day54	USA	MR ST and T abnormalities	Died 68th DOL	c.1076_1080delCCGTG (p.Ala359Glyfs*4) and c.80G > A (p.Gly27Asp)	#3

DCM: dilated cardiomyopathy; DOL: day of life; F = female; HT: heart transplantation; M = Male; MR: mitral regurgitation; PH: pulmonary hypertension; PFO: patent foramen ovale; RBBB: right bundle branch block; TR: tricuspid regurgitation; WES: whole exome sequencing.

## Data Availability

Not applicable.

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
