# Peer review of "Compound Heterozygous Missense Variants in RPL3L Genes Associated with Severe Forms of Dilated Cardiomyopathy: A Case Report and Literature Review"

_children, 2022, doi:10.3390/children9101495_

Round 1
Reviewer 1 Report
Das et al describe the case of an infant who showed a fulminant dilated cardiomyopathy (DCM) leading to severe acute heart failure (HF) requiring heart transplantation. Whole exome sequencing identified heterozygous variant for c. 151 G>A p. (A51T) and c.691 G>T p. (V231F) in RPL3L gene, classified as likely pathogenic.
The topic is interest and the paper well written, nonetheless it deserves some comments
1) The cardiac condition described in this case report is quite rare, I wonder whether data on HF/DCM prevalence in the first years of life are available.
2) Table 1 reported clinical features of already published infants with DCM linked to RPL3L gene. It would be interest to know if already described patients showed some clinical differences when compared to the new case reported by the authors, considering also that this was a newly described genetic variant.
3) In Fig 1 (E) pedigree image shows that both parents carried the RPL3L gene variant previously detected in the daughter, otherwise this data did not appear in the text.
4) I think an important message of the paper is the need of a genetic analysis in children and young patients diagnosed with DCM and/or HF. I think authors should emphasized this concept.
Author Response
Das et al describe the case of an infant who showed a fulminant dilated cardiomyopathy (DCM) leading to severe acute heart failure (HF) requiring heart transplantation. Whole exome sequencing identified heterozygous variant for c. 151 G>A p. (A51T) and c.691 G>T p. (V231F) in RPL3L gene, classified as likely pathogenic.
The topic is interest and the paper well written, nonetheless it deserves some comments
- Thank you.
1) The cardiac condition described in this case report is quite rare, I wonder whether data on HF/DCM prevalence in the first years of life are available.
- We revised the discussion with a beginning, "Dilated cardiomyopathy is the most common form of cardiomyopathy and accounts for approximately 55–60% of all childhood cardiomyopathies, and the majority are present before one year. According to the pediatric cardiomyopathy registry database, the incidence of DCM is 1.1 cases per 100 000 person-years, but the incidence was 8 times higher (8.3 cases per 100 000 person-years) in children diagnosed at <1 year of age.6 (added reference #6).
2) Table 1 reported clinical features of already published infants with DCM linked to RPL3L gene. It would be interest to know if already described patients showed some clinical differences when compared to the new case reported by the authors, considering also that this was a newly described genetic variant.
- We summarized the clinical presentations of all seven previous patients, and the common features among all the clinical curse is fatal unless heart transplantation and familial clustering. We added a sentence at the beginning of the conclusion, "Our case report has a similar clinical presentation to other previously reported cases and strongly supports the pathogenicity of the described RPL3L variants."
3) In Fig 1 (E) pedigree image shows that both parents carried the RPL3L gene variant previously detected in the daughter, otherwise this data did not appear in the text.
- Thank you. We added, "Her mother was positive for c.151 G>A p.(A51T), and her father was positive for c.691 G>T p.(V231 F). Both parents are heterozygous carriers (Figure 1E)."
4) I think an important message of the paper is the need of a genetic analysis in children and young patients diagnosed with DCM and/or HF. I think authors should emphasized this concept.
-We added at the end of the conclusion, "This case report emphasizes the importance of genetic testing, especially of WES for neonatal DCM, as it can identify variants of possible pathologic significance and provide an opportunity for early clinical management. The ability to promptly diagnose a neonate with a genetically caused DCM helps the clinicians discuss with the family various options, including heart transplantation."
Reviewer 2 Report
Bibhuti et al. present a case of fulminant HF in an infant, and postulate a causal role for compound heterozygous RPL3L variants. Reports on the causal role of RPL3L variants in neonatal DCM are sparse and include only 2 works, the latter by partially the same authorship as the presented manuscript. The evidence on the pathogenic role of RPL3L variants is not definitive, and although the presented case does not provide them either, it is valuable as it contributes to better documenting this association.
However, the authors should provide more detailed information:
- Technical parameters of WES shuld be provided such as mean coverage or % of targets covered <10x;
- Did you identify any other rare variants, especially in cardiomyopathy-related genes?
- Did you perform CNV analysis?
- Can you show which Criteria for Classifying Pathogenic Variants according to ACMG are met to classify the described variants as LP?
Other remarks:
- Gene names should be italicized
- Basic information on WES is unnecessary in the abstract
- The sentence "The ribosomal protein large 3-like (RPL3L) gene is a ribosomal protein.." has to be corrected (line 24)
- The discussion is for the most part a repetition of the discussion from the authors'previous work;
- The authors state (line 32): Three-dimensional homology modeling analysis of affected variants in RPL3L indicates that “they alter the interaction of RPL3L with the RNA components of the 60S ribosomal subunit and thus destabilize its binding to the 60S subunit; - It was done in the work by Ganapathi et al.
Can you perform in silico structural modeling of your identified variants or at least discuss their localization in the protein and potential influence on its function?
Author Response
Bibhuti et al. present a case of fulminant HF in an infant, and postulate a causal role for compound heterozygous RPL3L variants. Reports on the causal role of RPL3L variants in neonatal DCM are sparse and include only 2 works, the latter by partially the same authorship as the presented manuscript. The evidence on the pathogenic role of RPL3L variants is not definitive, and although the presented case does not provide them either, it is valuable as it contributes to better documenting this association.
- We added this in the revised manuscript how our previous two cases helped us to manage our 3rd case in a 6 months period. "our experience from the previous case report helped us to identify the present case to send the WES earlier and presented our case to our multidisciplinary team rigorous risk-benefit assessment. We successfully bridged the patient for heart transplant using Berlin Excor. We believe this is the tip of the iceberg, and unless tested for these rare possible pathologic variants, we will never identify the high-risk patients."
However, the authors should provide more detailed information:
- Technical parameters of WES should be provided such as mean coverage or % of targets covered <10x;
- We added in the revised text under discussion that for this gene, 100% of the coding region was covered at a minimum of 10x by this test. (Line 94-95). The quality metric is detailed in the report and includes: mean depth of coverage 197X and quality threshold 98.9% (this is not included in the report).
- Did you identify any other rare variants, especially in cardiomyopathy-related genes?
- We did not find any other variants. Other genetic tests for mitochondrial disorder by sequence analysis and deletion testing of the mitochondrial genome were negative.
- Did you perform CNV analysis?
- As per the report from GeneDx: the copy number variants (CNV) analysis was performed, and there is no indication of a clinically relevant deletion or duplication of three or more exons in the data for our patient. Added this to the case report section.
- Can you show which Criteria for Classifying Pathogenic Variants according to ACMG are met to classify the described variants as LP?
- Table-3 in ACGM report (ref#5) defines the criteria for pathogenic variant. We added, "ACMG has provided the evidence framework by the type of evidence as the strength of criteria for a benign or pathologic assertion of a variant. As per ACMG, our patient phenotype is highly specific for a disease with a single genetic etiology, and prior sources recently reported such variants as pathogenic.2, 3 The details of the classification of a sequence variant are beyond the scope of this paper, and readers could refer to www. clinivar.com for details."
Other remarks:
- Gene names should be italicized: Done
- Basic information on WES is unnecessary in the abstract: Edited and removed.
- The sentence "The ribosomal protein large 3-like (RPL3L) gene is a ribosomal protein.." has to be corrected (line 24): We agree with Reviewer but the RPL3L is described in full as reported by Ganapati et al.
- The discussion is for the most part a repetition of the discussion from the authors previous work: -We modified the discussion relevant to the present case and lessons learned from our first case report.
- The authors state (line 32): Three-dimensional homology modeling analysis of affected variants in RPL3L indicates that “they alter the interaction of RPL3L with the RNA components of the 60S ribosomal subunit and thus destabilize its binding to the 60S subunit; - It was done in the work by Ganapathi et al.
- We change the sentence to "All identified variants have been missense variants predicted to destabilize RPL3L binding to the 60S ribosomal subunit.2 "
Can you perform in silico structural modeling of your identified variants or at least discuss their localization in the protein and potential influence on its function?
- Ref: Table-2 in ACMG report. The following lines added, "A variety of in-silico tools are both publically and commercially available, which helps interpret sequence variants identified by WES. The details of the different software programs and algorithms for their prediction are beyond the scope of the present report. Interested readers can refer to the ACMG report for details.5"
Round 2
Reviewer 2 Report
The authors responded to some of my comments and provided a revised version of the manuscript. They presented the scope of the genetic test, partly in the case report (WES, CNV), partly in the discussion (study of the mitochondrial genome) - it requires ordering . For some reason they did not include the depth of coverage information in the text - it should be included (an amazing 197x). They did not answer the question what% of the exome (not RPL3L) was covered <10x - this information is useful to assess the risk of missing some causal variants.
Information that you have not identified other rare variants should be included in the case report, not in the discussion.
Both variants are VUS in the Clinvar - that's why I think the authors should write why they consider them LP. Moreover, they write in other places they are pathogenic. I think it is not appropriate to refer the readers to the general principles for classifying variants.
Also, the formulation "dilated cardiomyopathy ... associated with ... RPL3L gene pathologic variants" (line 17) is too strong as this relationship is not proven (i.a. because these variants are not pathogenic). The authors don't use in-silico models to predict the impact of the variant - I think they should not suggest readers can do it themselves. Neither they discuss the potential effects of the identified variants on protein function, based on eg. location of the substituted amino acid.
I think they should at least go into more detail on how other etiologies have been excluded. What metabolic and infectious workup was used in this case?
The authors used italics in variant names - it was unnecessary. Gene names should be italicized (i.e. RPL3L)
In conclusion, I believe that the manuscript requires ordering and, above all, a more detailed presentation of diagnostic methods aimed at detecting the causes of the disease (both genetic and other), in order to make the causal role of the identified variants (VUSes) as likely as possible (since no other evidence is available)
In conclusion, I believe that the manuscript requires ordering and, above all, a more detailed presentation of diagnostic methods aimed at detecting the causes of the disease (both genetic and other), in order to make the causal role of the identified variants as likely as possible (since no other evidence is available)
